# Activity-Based Exposure Levels and Cancer Risk Assessment Due to Naturally Occurring Asbestos for the Residents Near Abandoned Asbestos Mines in South Korea

**DOI:** 10.3390/ijerph18105225

**Published:** 2021-05-14

**Authors:** Seungho Lee, Dongmug Kang, Youngki Kim, Yoon-Ji Kim, Se-Yeong Kim

**Affiliations:** 1Department of Occupational and Environmental Medicine, Pusan National University Yangsan Hospital, Yangsan 50612, Korea; cjsfhleo10@pusan.ac.kr (S.L.); kangdm@pusan.ac.kr (D.K.); mungis@pusan.ac.kr (Y.K.); 2Department of Preventive and Occupational & Environmental Medicine, Medical College, Pusan National University, Yangsan 50612, Korea; harrypotter79@pusan.ac.kr; 3Environmental Health Center of Asbestos, Pusan National University Yangsan Hospital, Yangsan 50612, Korea

**Keywords:** naturally occurring asbestos, abandoned asbestos mine, activity-based sampling, excess lifetime cancer risk

## Abstract

This study aims to evaluate the overall asbestos exposure intensity and assess the health risk to residents due to naturally occurring asbestos (NOA) near abandoned asbestos mines in South Korea. Of 38 mines, we found 19 with measured concentrations of NOA. We evaluated the average of airborne NOA concentrations according to the environmental exposure category. When evaluated regionally by dividing into two clusters, the mean concentrations in activity-based sampling (ABS) scenarios exceeded the Korean exposure limit (0.01 f/cc) in both clusters. Moreover, airborne NOA concentrations in agricultural activity (5.49 × 10^−2^ f/cc) and daily activity (6.95 × 10^−2^ f/cc) had the highest values for clusters A and B, respectively. The excess lifetime cancer risk of one region (cluster A) by the ABS scenarios did not exceed the Korean Ministry of the Environment’s criteria for soil purification (1 × 10^−4^). However, one of the ABS scenarios—the daily life activity of clusters centered on Chungcheongbuk-do (cluster B)—showed an exposure of 1.08 × 10^−4^, greater than the limit (1 × 10^−4^). This indicates non negligible health damage to residents living near the abandoned asbestos mines, and it is necessary to continuously monitor and clean up the asbestos contamination.

## 1. Introduction

Asbestos generally refers to six regulated fibrous minerals—chrysotile, anthophyllite, actinolite, tremolite, amosite, and crocidolite. However, from a public health perspective, naturally occurring asbestos (NOA) refers to asbestos and asbestos-like minerals that are unintentionally mined or used, whose disturbance in the environment could cause exposure and possibly asbestos-related diseases (ARDs) [1,2]. Environmental exposure to NOA occurs due to a natural process or human activities in daily life settings, such as farming, transportation, and recreational exposures [1,2].

There are prior studies on health hazards due to environmental exposure, but not to occupational asbestos exposure. In Canada, it has been shown that the estimated lifetime mortality risk of lung cancer and mesothelioma combined varied between 1.4 and 4.9 per 100,000 persons continuously environmentally exposed to asbestos for 80 years in a mining town [3]. In addition, a study on the malignant mesothelioma of residents living near abandoned factories using asbestos in Japan showed a risk of environmental exposure [4]. Similarly, a study near an asbestos production plant in Egypt reported a high incidence of malignant mesothelioma due to environmental exposure and showed that many asbestos fibers were measured in the atmosphere, even after the plant was shut down [5]. In the province of Diyarbakir in Turkey, the incidence of malignant mesothelioma was higher in areas near NOA and aligned with the dominant wind direction [6]. Likewise, it was reported that environmental exposure to Libby vermiculite in Minnesota increased the incidence and mortality of ARDs, including malignant disease [7]. Numerous studies have also shown that there is a relationship between the onset of malignant mesothelioma and environmental asbestos (whether NOA or not) exposure [8,9,10,11,12]. These studies emphasized the health risk assessment to prevent ARDs due to environmental asbestos exposure.

In South Korea, asbestos mining began in the 1930s and was temporarily stopped after World War II in 1945. Mining commenced for the slate industry again from the 1950s and was stopped once more due to productivity problems in 1983. [13]. Even after asbestos mining was stopped in South Korea, abandoned asbestos mines have been considered a major environmental exposure sources of asbestos [14]. In an epidemiological survey of four towns in Chungcheongnam-do, which has the largest asbestos mine in Asia, more than 50 out of about 200 people were diagnosed with asbestosis. As most of the people diagnosed with asbestosis did not have a history of working in asbestos mines professionally, it is thought that their condition was due to environmental exposure [15]. In 2009, the Korean Ministry of the Environment (KME) determined that asbestos contamination was serious and initiated a survey of abandoned asbestos mines [16], followed by precision investigations and restoration projects. Subsequently, several studies have been published and reported in South Korea [17,18,19,20,21,22,23,24,25,26,27,28,29,30,31]. These studies measured the concentration of NOA and assessed attendant health risks. However, most of them only contained scattered data for a small number of subjects and defined data centered on a single mine. Therefore, they were limited for the evaluation of the overall extent of exposure to NOA near abandoned asbestos mines in South Korea. These limitations made it difficult to use the data in victim compensation policies for NOA exposure. To address these limitations, in this study, we collected all possible data of environmental NOA exposure near abandoned asbestos mines and assessed the overall value of NOA exposure in South Korea. Further, we evaluated the health risk of residents living near the abandoned asbestos mines with the excess lifetime Cancer Risk (ELCR).

## 2. Materials and Methods

### 2.1. Source of Data, Data Extraction

We conducted an integrated literature review of surveys covering the concentration of NOA exposure of residents living near abandoned asbestos mines in South Korea through a literature search in the Korean journal database and internet-based academic reports related to asbestos mines, asbestos-like mines, and NOA. We also collected data on abandoned asbestos mines in South Korea from the Asbestos Environmental Health Center of Yangsan Pusan National University Hospital and Soonchunhyang University Hospital and KME. We found 2194 articles from the search, of which 2158 were excluded on the basis of their titles and abstracts. There were 2103 irrelevant and 55 duplicated studies. Next, we additionally excluded 21 studies after reading the full texts. In this step, studies where asbestos concentrations were neither expressed in f/cc nor analyzed by transmission electron microscopy (TEM), or asbestos measurement did not follow the “2010 Soil Environment Management Guidelines for Asbestos Generating Area such as Asbestos Mine” of the KME [32], were excluded. These guidelines are based on the United States Environmental Protection Agency (USEPA) guide, covering asbestos-contaminated sites [33]. Finally, we short-listed 15 studies to be included in this pooled analysis. The procedure for literature selection is shown in Figure 1, and the characteristics of the studies selected are shown in Table 1.

Through the 15 selected studies, we identified the abandoned asbestos mines and their locations in South Korea. We then collected the measured concentration data of NOA near these mines. There were 38 mines with a history of mining activity in South Korea. We found that measurements of NOA concentrations were available for 19 of the 38 mines. NOA concentrations were measured within 4 km^2^ of the mine development area according to the KME guidelines. For 19 mines without airborne NOA concentration measurements, asbestos detection was within 0.25% of the soil in a basic survey, or there were no residential areas near the mine development area [32]. As mentioned, only the concentration values expressed in units of f/cc and analyzed via a TEM were extracted. There were two types of concentration data: one, a raw measurement, and the other measured using activity-based sampling (ABS). The ABS method was used to measure the concentration of asbestos scattered through the soil disturbance activity in a specific scenario and determine the effect of asbestos scattering on the human body, as in reference [34]. These asbestos exposure concentration scenarios included outdoor, indoor, and 11 ABS methods (bicycle, car, motorcycle, cultivator, walking, weeding, weed whacking, digging, field sweeping, physical training, and children playing in the dirt) suggested in the KME guidelines [32]. We extracted the information necessary to determine the frequency and duration of asbestos exposure activities among residents living near the abandoned asbestos mines. This was obtained from a survey of residents living near these mines and included daily activity (exposure) hours, annual activity (exposure) days, first activity (exposure) age, and duration of activity (exposure).

### 2.2. Calculation of Airborne NOA Concentration

All data were merged to calculate airborne NOA concentrations near the abandoned asbestos mines. When merging data, divisions according to measurement seasons were ignored to arrive at a large sample size of combined data. We grouped the 11 ABS methods into three categories (Table 2). The 11 ABS methods covered here are based on the guidelines of the KME [32]. These scenarios are almost similar to USEPA standard operating procedures. The type of scenario used is site-specific [35]. The results were expressed as the arithmetic means. The data expressed as the geometric means were included in the calculation after conversion to the arithmetic means, assuming a log-normal distribution. Next, we classified the mines into two, according to geopolitical location (areas near Chungcheongnam-do and Chungcheongbuk-do) and calculated each average.

### 2.3. ELCR

To assess the risk of residents living near the abandoned asbestos mine due to NOA, we calculated the ELCR, which is an estimation of the risk of developing cancer due to site-related exposure. ELCR can be used for determining whether airborne concentrations of asbestos are associated with unacceptable risks to human bodies at specific sites, which is based only on the prediction of excess cancer risk through respiratory exposure [33]. The ELCR formula given by the USEPA [33] is as follows:ELCR = EPC × TWF × IUR(1)
where exposure point concentration (EPC) is the concentration of asbestos fibers in the air (f/cc) for the specific activity being assessed, time weighting factor (TWF) is the factor accounting for less-than-continuous exposure during a one-year exposure, and inhalation unit risk (IUR) is a value of (f/cc)^−1^ given by the USEPA [33].
(2)$$\mathrm{TWF}\;=\;\frac{\mathrm{Exposure}\;\mathrm{Time}\;\left(\frac{\mathrm{hours}}{\mathrm{day}}\right)}{24}\times \frac{\mathrm{Exposure}\;\mathrm{Frequency}\;\left(\frac{\mathrm{days}}{\mathrm{years}}\right)}{365}$$

Exposure time corresponds to daily activity (exposure) hours, and exposure frequency corresponds to annual activity (exposure) days in the survey of residents living near the abandoned asbestos mines.

## 3. Results

### 3.1. Abandoned Asbestos Mines in South Korea

Based on the integrated literature review, there were 38 abandoned asbestos mines in South Korea with a history of asbestos mining and availability of an accurate location. Of these, 19 mines had measured concentrations of airborne NOA in the vicinity. We have mapped these 19 mines in the NOA geological map released by the KME in 2019. The map (Figure 2) shows a regional distribution that is likely to contain NOA on a 1:50,000 scale and was drawn through a review of geological evidence and literature data by the KME. Many abandoned asbestos mines were clustered in rock distribution areas with a high likelihood of NOA concentrations. Both amphibole and serpentine bearing rocks are found in Hongseong-gun, Chungcheongnam-do, whereas both metamorphic sedimentary rocks, including dolomite, siliceous limestone, and lime sedimentary rocks bearing rocks are found in Jecheon-si, Chungcheongbuk-do [36]. We classified these abandoned asbestos mines into two clusters, centered on Chungcheongnam-do (cluster A) and Chungcheongbuk-do (cluster B) (Table 3).

### 3.2. Airborne NOA Concentrations Near Abandoned Asbestos Mines in South Korea

We calculated the arithmetic mean of the airborne NOA concentration values of the two clusters (Figure 3). The airborne NOA concentration was classified into outdoor, indoor, transportation activity, agricultural activity, and daily life activity according to the type of environmental exposure. The mean airborne NOA concentration of agricultural activity (5.49 × 10^−2^ f/cc) was the highest in cluster A, whereas that of daily life activity (6.95 × 10^−2^ f/cc) was the highest in cluster B. The mean airborne NOA concentration of the ABS scenario (transportation/agricultural/daily life activity) exceeded the South Korean exposure limit (dash line, 0.01 f/cc) in both clusters [15]. The mean airborne NOA concentrations of cluster A were higher than those of cluster B, except for daily life activity (TEM).

### 3.3. Survey of Residents Near the Abandoned Asbestos Mines in South Korea

We calculated the average of the results of surveys of residents living near the abandoned asbestos mines through a pooled analysis. Similar to the concentration data, we classified the data (daily activity hours, annual activity days, first activity age, and duration of activity) into two clusters and calculated the TWF and IUR (Table 4). The values related to ABS are the average of the individual survey response values in the literature. Those surveys for which the number of respondents was not mentioned were excluded from the calculation process. There were no distinct differences between the two clusters. In the ABS scenarios, the TWF was ranked in descending order of transportation, agricultural, and daily life activities in both clusters.

### 3.4. Health Risk Assessment of Residents Near the Abandoned Asbestos Mines in South Korea

Using the above data, we calculated the ELCR to assess the health risk of residents living near the abandoned asbestos mines (Figure 4). As described above, ELCR can be calculated by multiplying the concentration, TWF, and IUR corresponding to each type of environmental asbestos exposure (transportation, agricultural, and daily life activities). The difference in ELCR between the two clusters showed similar results to those seen for NOA concentrations. The ELCRs of cluster A were higher than those of cluster B, except for daily life activity. The ELCR of agricultural activity (9.71 × 10^−5^) was highest in cluster A, whereas that of daily life activity (1.08 × 10^−4^) was highest in cluster B. The ELCR exceeded the KME’s criteria for soil purification (0.0001) for the daily life activity of cluster B [32].

## 4. Discussion

In this study, we identified abandoned asbestos mines in South Korea and calculated the average airborne NOA concentrations, divided into two representative regions (clusters A and B). The results showed that the mean airborne NOA concentration of several ABS scenario categories (transportation, agricultural, and daily life activities) in both clusters exceeded the environmental exposure limit (0.01 f/cc) in South Korea [15]. We also found that the airborne NOA concentration was higher in the ABS scenarios than the others (outdoor and indoor), which may be because much NOA was scattered when there was soil disturbance. The dispersion of asbestos in soil positively and negatively correlated with wind velocity and water content, respectively [37]. Therefore, the measured NOA concentration with human activities that cause soil disturbance was higher than the measured airborne NOA concentrations in the outdoor and indoor scenarios. There are few detailed prior studies on the difference in airborne NOA concentrations between ABS scenarios. We found that airborne NOA concentrations in agricultural and daily life activities in clusters A and B, respectively, had the highest values. It is speculated that agricultural or daily life activity may cause stronger soil disturbances than transportation activity [36]. When comparing the two clusters, the airborne NOA concentrations in cluster A were generally higher than those in cluster B, which may be due to differences in the mineral properties of asbestos and the mechanism of NOA formation in the clusters. According to the report from the KME, serpentine asbestos (chrysotile) and amphibole asbestos (tremolite and actinolite asbestos) occurred in the Hongseong-eup area of Hongseong-gun (cluster A). Whereas, in the Susan-myeon/Deoksan-myeon area of Jecheon-si (cluster B), amphibole asbestos (tremolite and actinolite asbestos) occurred in metamorphic sedimentary rocks. Serpentine rock has a higher probability of NOA occurrence than metasedimentary rock [36].

Considering global studies of environmental asbestos exposure to residents living near asbestos mines, the concentrations near the mines in Italy and France were 2.5 and 1–17 f/L, respectively. For Canadian mines, the concentration was 46 and 10 f/L in 1974 and 1984, respectively [38]. In the Thetford mine area—the world’s largest asbestos mine in Canada—from 1905 to 1965, an annual average of above 200 f/L was reported [39]. Comparing these concentrations with the results of this study, we found that outdoor results are similar to concentrations near asbestos mines in Italy and France. There are few reports of airborne NOA concentrations for the ABS scenarios used in this study near the abandoned asbestos mines in other countries. The airborne NOA concentrations measured during simulated agricultural activities near the abandoned asbestos mine in Balangero, Italy, ranged within 16–26 f/L with a maximum of 25–40 f/L [40].

Whether through occupational or environmental exposure, asbestos affects human health through inhalation. Inhaled asbestos fibers affect the human body through cellular and molecular effects, such as mechanical damage, the production of reactive oxygen and nitrogen species, change of cellular signal transduction, activation of oncogenes, loss of tumor suppressor genes, and DNA mutation [41]. The ABS guidelines from KME and USEPA consider only the effects of asbestos on the human body through inhalation [32,33]. Therefore, we calculated ELCR to assess the health risks for respiratory carcinogenicity of residents living near the abandoned asbestos mines. The highest ELCR values in the ABS scenarios were agricultural activity (9.71 × 10^−5^) in cluster A and daily life activity (1.08 × 10^−4^) in cluster B. ELCR refers to the number of cancer cases that are expected to additionally develop for a given number of people when exposed to a dose specified for carcinogen [42]. The reference value for the airborne NOA concentration was presented from the malignant risk level of 1 in 1,000,000 to 1 in 10,000 people [43]. When ELCR exceeds 0.0001, soil purification in that area is required according to KME [32]. In our results, the ELCR did not exceed this limit for transportation, agricultural, and daily life activities in cluster A. However, it exceeded the KME’s criteria for soil purification (0.0001) for daily life activity in cluster B. Considering this, the risk of carcinogenicity due to NOA near the abandoned asbestos mines is not negligible, and soil restoration work may be required in some areas in South Korea.

It was reported that the incidence rate of malignant mesothelioma was 0.89 per 1000 person-years for residents living near the abandoned asbestos mines in Hongseong-gun (the region in cluster A), Chungcheongnam-do, South Korea, based on the National Health Insurance Database from 2007 to 2018 [44]. The increased risk of mesothelioma in residents living near the asbestos mines has been reported in other countries [7]. In a study of 4768 residents of the Wittenoom town in Australia, the hazard ratio was reported to be 3.88 among those whose age at first residence ≥15 years [45]. Although it was not limited to the vicinity of the mine, in a population-based case-control study in France, the non-occupational asbestos population-attributable risks of pleural mesothelioma were 20.0% (99%; CI: 33.5–73.5%) for men and 38.7% (99%; CI: 8.4–69.0%) for women [46].

As the first limitation of this study, it is difficult to generalize our results to calculations of all abandoned asbestos mines in South Korea. We could only use airborne NOA concentration measurements from 19 of the 38 identified abandoned asbestos mines. However, the mines without measurements were excluded from the precision investigation because the risk of exposure to NOA was low or there was no residential area near it, according to KME guidelines. Therefore, the pooled analyzed measurements are somewhat representative of the airborne NOA exposure concentration near the South Korean abandoned asbestos mines with risk of environmental NOA exposure. The second limitation is that we ignored the individual characteristics of each measurement data (weather, season, etc.) and differences in geological properties between the two clusters. The weather or season in which the airborne NOA concentrations were measured is relevant because the degree of scattering of asbestos in the soil is affected by the wind velocity and humidity in the soil. Similarly, the specific ABS results cannot be generalized to other scenarios, as environmental exposure concentrations due to asbestos depend on both the asbestos soil content in the area and activities performed on it [33]. However, in this study, we tried to ascertain and present the overall intensity of exposure to NOA near the abandoned asbestos mines in South Korea. Therefore, we had to include as much measurement data as possible. The third limitation is that ELCR calculated in this study was simplified; hence, it was difficult to apply to other situations. It was applicable only to specially defined ABS scenario settings. In addition, it did not consider the differences among adults, children, and adolescents, which affected IUR and height of exposure to dispersed NOA. Thus, it is difficult to determine which ABS scenario category is more dangerous by comparing the ELCR in this study. To accurately estimate asbestos exposure considering this aspect, gathering data on NOA in various ABS scenarios [47] and individual history of asbestos exposure is necessary.

Despite these limitations, we attempted to collect almost available data related to environmental asbestos exposure intensity near the abandoned asbestos mines in South Korea. Since 2011, the KME has been compensating environmental asbestos victims by implementing the asbestos damage relief system. The results of this study will help to evaluate the overall asbestos exposure intensity and assess the health risk to residents due to NOA near abandoned asbestos mines in South Korea. 

## 5. Conclusions

We calculated the average airborne NOA concentrations by collecting as much published data as possible and evaluated the health risk via ELCR in South Korea. The results indicated that the estimated exposure measurements via the ABS method of NOA exceeded the current environmental exposure limits in South Korea, and health damage to residents living near the abandoned asbestos mines was not negligible. Therefore, there is a requirement for continuous monitoring and cleanup work for asbestos contamination near the abandoned asbestos mines.

## Figures and Tables

**Figure 1 ijerph-18-05225-f001:**
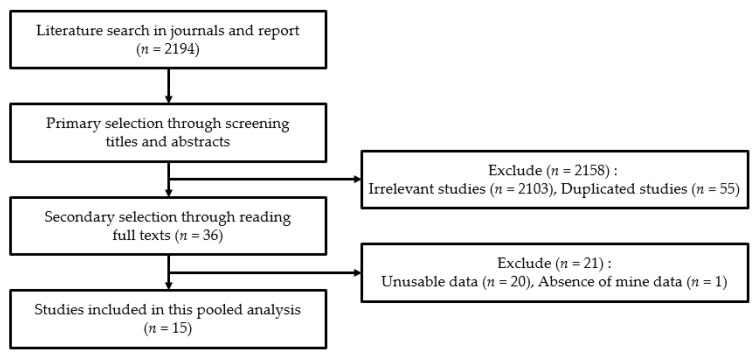
Literature selection procedure.

**Figure 2 ijerph-18-05225-f002:**
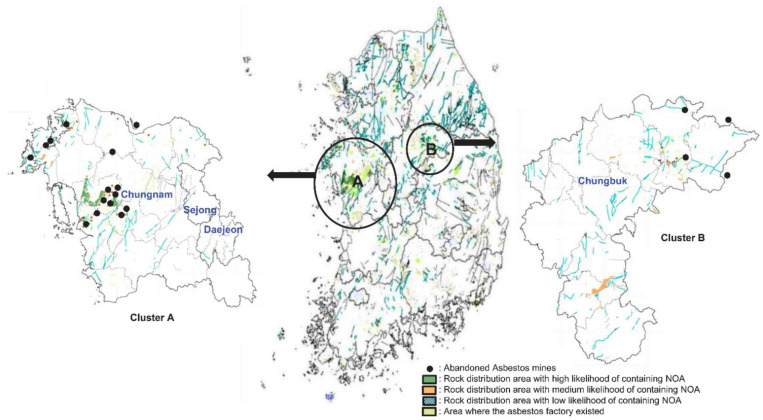
Location map of abandoned asbestos mines included in the pooled analysis in South Korea. (Map source: Asbestos Management Comprehensive Information Network; https://asbestos.me.go.kr, accessed on 22 April 2021).

**Figure 3 ijerph-18-05225-f003:**
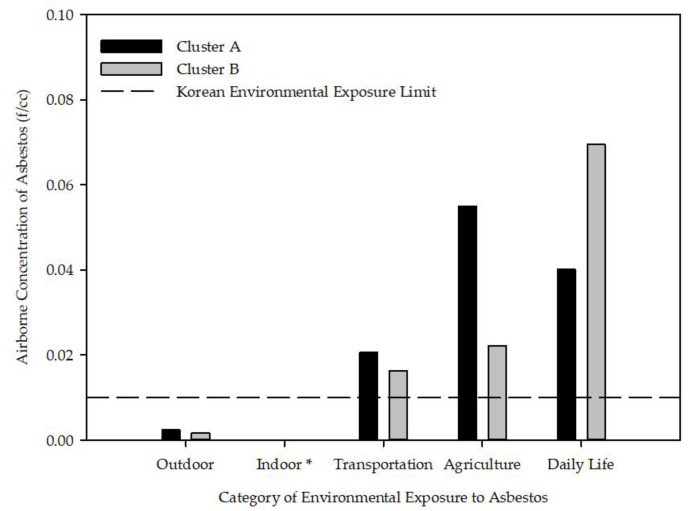
Airborne NOA concentrations of the abandoned asbestos mines in South Korea. * Airborne NOA concentration for indoor was too low to show in this figure (<0.0001 f/cc).

**Figure 4 ijerph-18-05225-f004:**
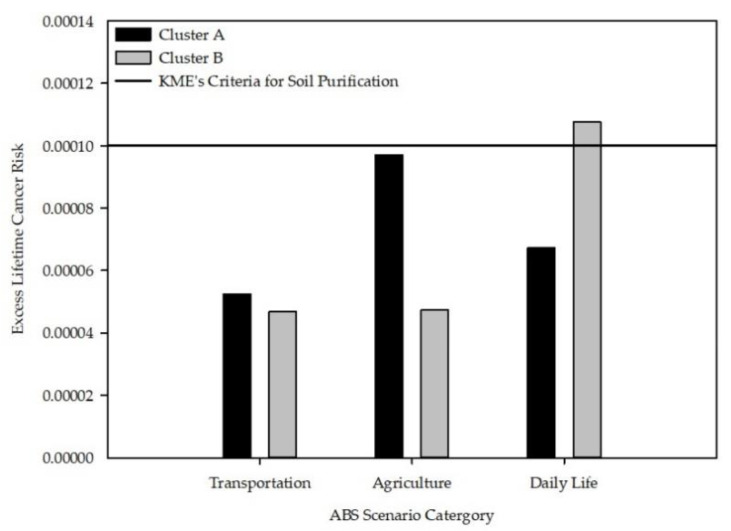
ELCR for health risk assessment of residents near abandoned asbestos mines in South Korea.

**Table 1 ijerph-18-05225-t001:** The characteristics of studies included in the pooled analysis.

No.	Author or Publisher/Year	Mine	Environmental Exposure Type	Survey *
1	Kim et al., 2009 [17]	Gwang-Cheon 1	Outdoor, Walking	No
2	Korean Ministry of the Envrionment, 2010a [18]	Gwang-Cheon 1, Sin-Seog, Dong-A	Outdoor, Indoor, Bicycle, Motorcycle, Car, Cultivator, Walking, Weeding, Weed whacking, Digging, Field sweeping, Physical training, Children playing in the dirt	Yes
3	Korean Ministry of the Envrionment, 2010b [19]	Dong-A, Hong-Dong-Baek-Dong, Hong-Dong-Gwang-Si	Outdoor, Indoor, Bicycle, Motorcycle, Cultivator, Walking, Weeding, Weed whacking, Digging, Field sweeping, Physical training,	Yes
4	Korean Ministry of the Envrionment, 2011 [20]	Bi-Bong, Yang-Sa, Shin-Deok	Outdoor, Bicycle, Motorcycle, Car, Cultivator, Walking, Weeding, Weed whacking, Digging, Field sweeping, Physical training,	Yes
5	Shin et al., 2011 [21]	Dong-A	Outdoor	Yes
6	Yoon, 2011 [22]	Gwang-Cheon 1, Sin-Seog, Dong-A	Outdoor, Motorcycle, Cultivator, Walking, Weeding, Digging, Field sweeping	Yes
7	Korean Ministry of the Envrionment, 2012 [23]	Bong-Hyun, Wol-Rim, Dae-Heung-Hong-Dong, Sin-Gok-Ri, Hyun-Deok	Outdoor, Indoor, Bicycle, Motorcycle, Car, Cultivator, Walking, Weeding, Weed whacking, Digging, Field sweeping, Physical training, Children playing in the dirt	Yes
8	Shin et al., 2012a [24]	Dong-A	Outdoor, Bicycle, Car, Weeding, Weed whacking, Field sweep	Yes
9	Shin et al., 2012b [25]	Dong-A, Shin-Deok	Outdoor, Bicycle, Car, Weeding, Weed whacking, Digging, Field sweeping	Yes
10	Korean Ministry of the Envrionment, 2013 [26]	Cheong-San-Ri, Dae-Cheon-Ri, Hong-Seong	Outdoor, Indoor, Bicycle, Motorcycle, Car, Cultivator, Walking, Weeding, Weed whacking, Digging, Field sweeping	Yes
11	Korean Ministry of the Envrionment, 2014 [27]	Gwang-Cheon 2, Jae-Jeong	Outdoor, Indoor, Bicycle, Motorcycle, Car, Cultivator, Walking, Weeding, Weed whacking, Digging, Field sweeping	Yes
12	Korean Ministry of the Envrionment, 2015 [28]	Young-Jin, Sin-Seon	Outdoor, Indoor, Bicycle, Motorcycle, Car, Cultivator, Walking, Weeding, Weed whacking, Digging, Field sweeping, Physical training	Yes
13	Lee et al., 2015 [29]	Gwang-Cheon 1, Sin-Seog	Outdoor, Bicycle, Motorcycle, Car, Cultivator, Walking, Weeding, Weed whacking, Digging, Field sweeping, Physical training, Children playing in the dirt	Yes
14	Geum-River Basin Environmental Office, 2018 [30]	Sin-Seog	Outdoor, Indoor, Bicycle, Motorcycle, Cultivator, Walking, Weeding, Digging, Field sweeping	Yes
15	Wonju Regional Environmental Office, 2019 [31]	I-Wha	Outdoor, Indoor, Bicycle, Motorcycle, Car, Cultivator, Waling, Weeding, Weed whacking, Digging, Field sweeping, Physical training	Yes

***** Survey including daily activity (exposure) hours, annual activity (exposure) days, first activity (exposure) age, and duration of activity (exposure) of residents living near abandoned asbestos mines in South Korea.

**Table 2 ijerph-18-05225-t002:** ABS method classification in this study.

ABS Scenario Category	ABS Method
Transportation Activity	Bicycle, Motorcycle, Car, Cultivator, Walking
Agricultural Activity	Weeding, Weed whacking, Digging
Daily Life Activity	Field sweeping, Physical training, Children playing in the dirt

**Table 3 ijerph-18-05225-t003:** Location and number of survey respondents of the abandoned asbestos mines in South Korea.

Cluster	Administrative District		Asbestos Mine	Number of Survey Respondents		
	Province (do)	City (Si), County (Gun)		ABS Scenario Category		
				Transportation Activity	Agricultural Activity	Daily Life Activity
A	Chungcheongnam-do	Boryeong-si	Jae-Jeong	34	27	16
			Sin-Seog	163	177	86
		Cheongyang-gun	Bi-Bong Yang-Sa	104	81	26
		Godeok-myeon	Dae-Cheon-Ri	86	71	13
		Hongseong-gun	Dae-Heung-Hong-Dong Wol-Rim	111	82	50
			Gwang-Cheon1	216	203	157
			Hong-Dong-Baek-Dong	36	63	61
			Hong-Seong	86	40	13
		Seosan-si	Gwang-Cheon2	34	27	16
		Taean-gun	Cheong-San-Ri	75	52	14
			Shin-Deok	91	73	26
			Young-Jin	86	40	13
	Gyeonggi-do	Pyeongtaek-si	Hyun-Deok	67	45	30
B	Chungcheongbuk-do	Jecheon-si	Dong-A	192	276	113
			Sin-Seon	115	82	31
	Gangwon-do	Yeongwol-gun	I-Wha	Not mentioned in the literature *****		
	Gyeongsangbuk-do	Yeongju-si	Bong-Hyun	93	61	29

***** Number of survey respondents was not found in the literature.

**Table 4 ijerph-18-05225-t004:** TWF and IUR based on survey results of residents near the abandoned asbestos mines in South Korea.

Cluster		ABS Scenario Category					
		Transportation Activity (Value/*n* *)		Agricultural Activity (Value/*n* *)		Daily Life Activity (Value/*n* *)	
A	TWF	0.04	1247	0.03	1095	0.02	568
	IUR	0.057		0.057		0.083	
B	TWF	0.05	400	0.04	419	0.02	173
	IUR	0.057		0.06		0.069	

* = number of respondents.

## Data Availability

The data of published paper used in this study is available from reference [21,22,24,25,29]. The data of report is available from the Korean Ministry of the Environment (KME). However, this data is available with the approval of the KME.

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
