# Peer review of "Activity-Based Exposure Levels and Cancer Risk Assessment Due to Naturally Occurring Asbestos for the Residents Near Abandoned Asbestos Mines in South Korea"

_ijerph, 2021, doi:10.3390/ijerph18105225_

Round 1
Reviewer 1 Report
The manuscript by Lee et al. entitled 'Activity Based Exposure Levels and Cancer Risk Assessment due to Naturally Occurring Asbestos for the Residents Around Abandoned Asbestos Mines in South Korea' is an interesting study regarding potential risks of environmental asbestos exposure. I have one minor comment for the authors: Figure 2 indicates a green diamond as the site of the prior Asbestos factory. I can't see it in the figure. Please add the diamond or make it more apparent on the image.
Reviewer 2 Report
this paper need extensive alterations if it is to be published. There are many areas of the world eg. Canada,India where mining communities have been studied and should be cited. The introduction is woefully incorrect or shows a lack of understanding. Asbestos, as generally define is a group of six regulated fibers, one serpentine and 5 amphiboles. NOA should ONLY refer to these fibers and NOT to "fibrous minerals.....rocks and soils". These are NOT asbestos but something else. I could not access (without paying) reference 5 but cannot imagine that any NOA anywhere is more harmful than regulated asbestos(which as used here is redundant). Reference 6 speaks to exposure, which is ubiquitous everywhere in the world, so more relevant is a reference or two about disease near a facility, not exposure(eg Kubota plant in Japan, cement factory in Egypt etc.). Line 46 speaks to "accurate exposure assessment" which is really false and misleading. There is never individual quantitative assessment and any such measurements are approximations. One really is talking about qualitative assessment, not quantitative and this should be made clear. As used in 2nd full paragraph of page 2 the term asbestos here, correctly, refers to regulated asbestos fibers. 15 articles covering only 19 of 38 mines an incomplete snapshot and generalized calculations about risk are not really warrented. Lines 157/8 speak to"primary types"- what else is there. There is no secondary types of true asbestos than the six fibers discusses. Figure 3 seems odd-there should really have been some asbestos found indoors in such settings. For figure 3 and 4 the idea of TOTAL makes no sense lumping the two clusters together,especially with so may mines not represented. The calculations of risk are ,again, only relevant to these settings and may or may not have any bearing on a wider set of mines or situations. The discussion leaves out much it should include. The fact that only some mines have data, and differences even with these two clusters, makes the finds not at all generalizable. Data about risk, such as the work of LaCourt should be cited to show that the levels calculated here carry a high risk. If available, some data about the actual rates of diseaee, esecially mesotheliomas, in these mining communities would be helpful. There is so much to correct on this paper.
Reviewer 3 Report
- The authors only showed the location of the abandoned asbestos mine, and did not point out the location of the sampling point, and the distance from the sampling point to the abandoned asbestos mine and other relevant detail information.
- What is the way asbestos fibre affects human health?
- The index EPC is the exposure concentration in the observation area, but from this manuscript, the authors does not calculate the exposure concentration of each grid area shown in Figure 2.
- How reliable is the results of the authors' health risk assessment in this manuscript? How to verify it?
- Other suggestions,
Figure 2 is not clear and needs to be redrawn in professional software.
The author needs to modify the language of the manuscript
Line 66-67, please add related Literature
Round 2
Reviewer 2 Report
Appreciate the corrections made. Line 159 still needs to be changed. These are not the MAJOR TYPES, they are the only types. Simply put- this could simply say that both amphibole and serpentine bearing rock are found .......
Reviewer 3 Report
I have no other suggestions.
